# Development of an integrated model framework for multi-air-pollutant exposure assessments in high-density cities

Zhiyuan Li[1,2], Kin-Fai Ho[3], Harry Fung Lee[4], Steve Hung Lam Yim[5,6,7,*]

[1]School of Public Health (Shenzhen), Sun Yat-sen University, Guangzhou 510275, China
[2]Institute of Environment, Energy and Sustainability, The Chinese University of Hong Kong, Shatin, N.T., Hong Kong, China
[3]The Jockey Club School of Public Health and Primary Care, The Chinese University of Hong Kong, Shatin, N.T., Hong Kong, China
[4]Department of Geography and Resource Management, The Chinese University of Hong Kong, Shatin, N.T., Hong Kong, China
[5]Asian School of the Environment, Nanyang Technological University, Singapore
[6]Lee Kong Chian School of Medicine, Nanyang Technological University, Singapore
[7]Earth Observatory of Singapore, Nanyang Technological University, Singapore

*Correspondence to*: Steve Hung Lam Yim (steve.yim@ntu.edu.sg)

**Abstract.** Exposure models for some criteria air pollutants have been intensively developed in past research; multi-air-pollutant exposure models, especially for particulate chemical species, have been however overlooked in Asia. Lack of an integrated model framework to calculate multi-air-pollutant exposure has hindered the combined exposure assessment and the corresponding health assessment. This work applied the land-use regression (LUR) approach to develop an integrated model framework to estimate 2017 annual-average exposures of multiple air pollutants in a typical high-rise and high-density Asian city (Hong Kong, China) including four criteria gaseous air pollutants [particulate matters with an aerodynamic diameter equal to or less than 10 µm ($PM_{10}$) and 2.5 µm ($PM_{2.5}$), nitrogen dioxide ($NO_2$), and ozone ($O_3$)], as well as four major $PM_{10}$ chemical species. Our integrated multi-air-pollutant exposure model framework is capable of explaining 91–97% of the variability of measured gaseous air pollutant concentration, with the leave-one-out cross-validation $R^2$ values ranging from 0.73 to 0.93. Using the model framework, the spatial distribution of the concentration of various air pollutants at a spatial resolution of 500 m was generated. The LUR model-derived spatial distribution maps revealed weak to moderate spatial correlations between the $PM_{10}$ chemical species and the criteria air pollutants, which may help to distinguish their independent chronic health effects. In addition, further improvements in the development of air pollution exposure models are discussed. This study proposed an integrated model framework for estimating multi-air-pollutant exposure in high-density and high-rise urban areas, serving an important tool for multi-air-pollutant exposure assessment in epidemiological studies.

# 1 Introduction

Ambient air pollution has been identified as one of the most important health risk factors, contributing to premature deaths and disabilities worldwide (Bowe et al., 2018; Burnett et al., 2018; HEI, 2019; Yim et al., 2019; Yim et al., 2022). In 2017, air pollution was ranked as the fifth among all-mortality risk factors globally, accounting for nearly 5 million premature deaths (HEI, 2019). Ambient $PM_{2.5}$ (particles of aerodynamic diameter less than or equal to 2.5 μm) was associated with 2.9 million premature deaths, and ozone ($O_3$) accounted for approximately 0.5 million premature deaths in 2017 (HEI, 2019). Numerous previous epidemiological studies have documented good evidence of the positive association between air pollution exposures and various types of health-effect endpoints, such as stroke, heart diseases, asthma, and lung cancer (Crouse et al., 2015; Fan et al., 2018; Renzi et al., 2019; Wang et al., 2017; Xue et al., 2021). For example, Renzi et al. (2019) estimated that there were increases of 0.8%, 0.9%, and 1.4% in non-accidental, cardiovascular, and respiratory mortality, respectively, for every 1 μg/m³ increase in annual-average $PM_{10}$ (particles of aerodynamic diameter less than or equal to 10 μm) concentration in the Latium region of Italy during 2006–2012.

Polluted air mass contains a complex mixture of toxic particles and various gas-phase pollutants. Health effects from air pollution are consequences from combined exposure to air pollution mixtures (Coker et al., 2016; Levy et al., 2014; Stafoggia et al., 2017; Wang et al., 2022a; Xue et al., 2021; Yim et al., 2022). For instance, Wang et al. (2022a) found that $PM_{10}$ and $O_3$ dominated the health effects of air pollution mixtures on obstructive sleep apnea, a common sleep-related breathing disorder, in a cross-sectional study in Beijing, China. Up until now, most epidemiological studies targeting at air pollution and various health endpoints have typically focused on estimating the adverse health effects associated with exposure to a single air pollutant (or pollutant category) mainly due to the difficulties of conducting a multi-air-pollutant exposure assessment (Dominici et al., 2010; Wang et al., 2022b). In recent years, the scientific community has been moving toward a multi-air-pollutant concept to quantify the health hazards of air pollution mixtures as a whole (Chen et al., 2020a; Dominici et al., 2010; Mauderly et al., 2010; Vedal et al., 2010; Xue et al., 2021; Yim et al., 2022). To achieve this target, we need to work on multi-air-pollutant exposure assessment to support the corresponding health-related studies (Billionnet et al., 2012; Mauderly et al., 2010).

In order to assess health effects of air pollution mixture, an integrated model framework is urgently needed to estimate exposures to multiple air pollutants. However, the available exposure assessment studies typically focused on one or several criteria air pollutants, mainly traffic-related air pollutants including $PM_{2.5}$ and $NO_2$ (Cai et al., 2020; Cordioli et al., 2017; Hoek et al., 2008; Jin et al., 2019; Luminati et al., 2021; Ross et al., 2007; Xu et al., 2019). For example, Jin et al. (2019) estimated annual average exposures to $PM_{2.5}$ and $NO_2$ in Lanzhou, China, with $R^2$ values at 0.77 and 0.71, respectively. In addition, Luminati et al. (2021) developed a $NO_2$ exposure model in Sao Paulo, Brazil using the land-use regression (LUR) approach. It should be noted that chemical species of ambient particles has their significance but also independent toxicity and health risks (Li et al., 2022; Rappazzo et al., 2021; Requia et al., 2019; Wang et al., 2022b). Apart from the criteria

gaseous air pollutants, the chemical species of ambient particles should also be studied (Li et al., 2022; Rappazzo et al., 2021; Requia et al., 2019; Wang et al., 2022b). To the best of our knowledge, none of these previous studies has comprehensively evaluated the spatial heterogeneity among a large set of air pollutants (e.g., particulates and their chemical species, and gaseous pollutants) (Cai et al., 2020; Hoek et al., 2008; Li et al., 2021). Thus, it is essential to explore the establishment of an integrated multi-air-pollutant model framework to support epidemiological studies to isolate the health effects of multiple air pollutants.

The major objective of this study was to develop an integrated model framework for multi-air-pollutant exposure assessments in high-density and high-rise cities. The case of Hong Kong was illustrated to estimate annual-average exposures of major chemical species of ambient $PM_{10}$ as well as ambient $PM_{10}$, $PM_{2.5}$, $NO_2$, and $O_3$. Materials and methods, including the development and application of an integrated multi-air-pollutant model framework in Hong Kong, are described in Section 2. Section 3 presents the established multi-air-pollutant models and the spatial distribution maps of targeted air pollutants derived from the established models. The discussion and implications are provided in Section 4.

## 2 Materials and methods

We developed an integrated model framework for establishing multi-air-pollutant exposure models with two major modules of particulate matters (PM module) and gaseous pollutants (GAS module) (Figure 1). The PM and GAS modules were separated because the measurement and LUR modeling of PM species and gaseous pollutants are largely different in terms of measurement techniques, the number of required measurement sites, and selected predictor variables, etc. The integrated model framework handles the input datasets required for the PM and GAS modules and develops the LUR model for each air pollutant independently. The LUR models and the corresponding spatial distribution maps within each module can be used to further validate the LUR models and the corresponding spatial distribution maps under the same or another module (Li et al., 2021). For instance, in high-density cities, the spatial distribution of $O_3$ typically shows a generally opposite spatial variability compared with traffic-related air pollutants i.e., $NO_x$ because of the $NO_x$ titration. The established LUR models of the PM and GAS modules were used for assessing exposures to different air pollutants in epidemiological studies. In the present study, the PM module includes the LUR models for different sizes of PM and its chemical components, whereas the GAS module includes the LUR models for two typical gaseous pollutants of $NO_2$ and $O_3$. The included air pollutants can vary depending on the data availability when the proposed integrated model framework is applied for other cities in future studies.

### 2.1 Study area

Hong Kong (latitude 22°08'N and 22°35'N, longitude 113°49'E and 114°31'E) is a mountainous high-density and high-rise city situated at the southeast coast of the Pearl River Delta (PRD) region, China (Yim et al., 2009). Hong Kong has a total land area of about 1100 $km^2$, of which 24% is built-up area. It has a population of over 7 million, and the population density

of 6,690 people per square kilometer is among the highest in the world (Li et al., 2018). Hong Kong is characterized by cool and dry winters with an average temperature of 19 °C, and hot and humid summers with afternoon temperatures often above 31 °C and night temperatures around 26 °C (HKO, 2020). Hong Kong is typically influenced by the transboundary air pollution issue, when polluted air masses are transported from the PRD region and beyond to the region in winter (Lee et al., 2017; Li et al., 2020; Yim, 2020). The traffic density in Hong Kong is among the highest in the world, with 839,882 registered vehicles on 2100 km of road in 2017 (HKTD, 2020). Therefore, the traffic emission from different types of vehicles is another important source of air pollution in Hong Kong (Li et al., 2022).

## 2.2 Air pollution monitoring data

The Hong Kong air quality monitoring network provides a suitable demonstration because the locations of air quality monitoring stations (AQMSs) were chosen with reference to international guidelines and with practical consideration for the localized city characteristics (Figure S1 in the Supplementary Information). The environmental characteristics of the AQMSs are summarized in Table S1. The daily average concentration data of $PM_{10}$, $PM_{2.5}$, $NO_2$, and $O_3$, measured at 16 AQMSs, operated by the Hong Kong Environmental Protection Department (HKEPD), were collected from 1 January 2017 to 31 December 2017 (https://cd.epic.epd.gov.hk/EPICDI/air/station/, last accessed on September 2023). These AQMSs are generally diverse and representative, ranging from rural stations under a limited influence of anthropogenic emissions to traffic stations near the major roads in Hong Kong. $PM_{10}$ and $PM_{2.5}$ were measured continuously by automatic monitors, while the Opsis AR 500 system and the T-API 400 system were used to measure $NO_2$ and $O_3$ concentration, respectively (HKEPD, 2018). The details for the list of equipment for measurement of air pollutant concentration as well as the quality control and assurance procedures are documented in HKEPD (2018). All of the air pollutant concentration data had at least 345 daily values, which represented a relatively complete set of data. Due to data availability, the 2017 annual-average concentration of these air pollutants was estimated for development of the LUR models using collected daily air pollutant concentration data. In addition, the annual-average concentration of four major $PM_{10}$ chemical species, including total carbon (TC), nitrate ($NO_3^-$), sulfate ($SO_4^{2-}$), and cadmium (Cd), at 10 AQMSs was collected from the air quality reports of the HKEPD (https://www.aqhi.gov.hk/en/download/air-quality-reportse469.html?showall=&start=1, last accessed on September 2023) for development of the LUR models.

## 2.3 Potential predictor variables

All of the potential predictor variables with their corresponding buffer sizes and data sources are summarized in Table S2 and Figure S2. Meteorological variables (i.e., wind speed, wind direction, relative humidity, and temperature) were collected from nearby weather stations operated by the Hong Kong Observatory (https://www.hko.gov.hk/en/index.html, last accessed on September 2023). In addition, five categories of geospatial predictor variables including land use, road networks with traffic volume information, population density data, topography, and urban/building morphology were collected from various databases (Table S2). The ArcGIS software, version 10.6 (ESRI Inc., Redlands, CA, USA) was used to process these

datasets. The land-use type was classified into 10 main categories and 27 subcategories. The 10 main categories of land use included residential, commercial, industrial, institutional/open space, transportation, other urban or built-up land, agriculture, woodland/shrubland/grassland/wetland, barren land, and water bodies (https://www.pland.gov.hk/pland_en/info_serv/open_data/landu/index.html, last accessed on September 2023). The traffic volume data for different vehicle types were provided by the Hong Kong Transport Department (https://data.gov.hk/en-data/dataset/hk-td-tis_15-road-network-v2, last accessed on September 2023). Seven vehicle types, including private cars, non-franchised buses, light goods vehicles, franchised buses, medium and heavy goods vehicles, taxis, and public light buses were counted. Values of these geospatial variables in buffer sizes of 50 m, 100 m, 300 m, 500 m, 700 m, 1000 m, 2000 m, 3000 m, 4000m, and 5000 m around the AQMSs were estimated as the potential predictor variables using an ArcGIS buffer analysis. The geo-locations (longitude and latitude) were also adopted because they can reveal a north-south or west-east variability gradient of air pollutant concentrations that cannot captured by the selected predictor variables in the model (Huang et al., 2017). The geographical coordinate information for each station was obtained from the HKEPD.

## 2.4 Development of multi-air-pollutant exposure models

The LUR approach was based on the principle that air pollutant concentration at a given location depends on the environmental features (e.g., land-use types, traffic volume, meteorological conditions, etc.) of the surrounding area (Supplementary Text S1) (Li et al., 2022; Lu et al., 2020; Meng et al., 2015; Naughton et al., 2018; Wu et al., 2017). The supervised forward linear regression method was utilized to conduct the LUR modeling of multiple air pollutants (Eeftens et al., 2012; Eeftens et al., 2016; Huang et al., 2017; Jin et al., 2019; Liu et al., 2016; Saha et al., 2020). The method computed the direction of effect for a predictor variable to reflect the effect of the predictor variable on air pollutant concentration. It should be noted that the direction of effect can be positive or negative. Hence, the method first judged the direction of effect for each type of predictor variable based on the currently known relationship between the predictor variable and the corresponding air pollutant. As a secondary pollutant, $O_3$ is involved in many complex chemical reactions, and the expected directions of its effects were not as clear as that of other air pollutants (Li et al., 2022; Wolf et al., 2017), and $O_3$ has to be thus considered carefully in the process. Using the full dataset, we ranked all the predictor variables based on their adjusted explained variance (adjusted $R^2$) with air pollutant concentration. The predictor variable with the highest adjusted $R^2$ was selected to be included in the model when the direction of effect was consistent with our judgement. We then evaluated which of the remaining predictor variables further improved the adjusted $R^2$ of the LUR model, and selected the one giving the largest gain in the adjusted $R^2$ of the model and with the expected direction of effect. Subsequent predictor variables were not selected when they changed the direction of effect of one of the previously included predictor variables. This process continued to be proceeded until there were no more predictor variables with the expected direction of effect, which added at least 1% to the adjusted $R^2$ of the previous LUR model. Finally, the predictor variables with a $P$-value above 0.10 were removed from the LUR model. If the variance inflation factor (VIF), which measured the severity of multicollinearity in the regression analysis, was higher than 5.0, the predictor variable with the highest VIF was removed, and the model was then

re-established (Gulliver et al., 2018; Hsu et al., 2018; Jones et al., 2020; Ma et al., 2019; Zhang et al., 2015). Due to this procedure, the included predictor variables may obscure the potential influence of others.

The spatial autocorrelation (Moran's $I$) tool measured spatial autocorrelation using both feature locations and feature values simultaneously. Meanwhile, $z$-score and $P$-values were calculated to evaluate the significance of Moran's $I$ value. z-score values are standard deviations, whereas the Moran's $I$ index is bounded by -1.0 and 1.0. When the $z$-score or $P$-value indicates statistical significance, a positive Moran's $I$ index value indicates tendency towards clustering, whereas a negative Moran's $I$ index value indicates tendency towards dispersion (Cordioli et al., 2017; Luminati et al., 2021). Moran's $I$ index and the corresponding $z$-score and $P$-value on concentration residuals of the final LUR models were quantified using the ArcGIS software to evaluate their spatial autocorrelation. Leave-one-out cross-validation (LOOCV) was used to evaluate the predictive ability of the established LUR model to a new dataset (Ma et al., 2019; Wu et al., 2017). In brief, each station was withheld from the model sequentially, whereas the remaining stations were used to establish the model. The concentration at the withheld station was estimated using the established model in each iteration. The procedure was repeated until all the stations have been predicted once (Eeftens et al., 2016; Ji et al., 2019; Wolf et al., 2017). We validated the model performance using training and LOOCV $R^2$ values calculated based on the linear regression between measured and predicted concentration of the omitted stations. The statistical analysis was performed using the R statistical software, version 3.5.2 for Windows (R Foundation for Statistical Computing, Vienna, Austria).

**2.5 Spatial mapping of studied air pollutants**

The spatial distribution maps of predicted annual-average concentration of $PM_{10}$ and its chemical species, $PM_{2.5}$, $NO_2$, and $O_3$ were generated by our final LUR models, following the typical procedures of previous studies (Cai et al., 2020; Huang et al., 2017; Xu et al., 2019). A spatial resolution of 500 m was adopted here due to the spatial resolution of most predictor variables was at a spatial resolution of several hundred meters. The study area of Hong Kong was divided into grids at the spatial resolution of 500 m, and the air pollutant concentration at the centroid of each grid was estimated. Finally, the pollution distribution maps were generated using the predicted concentration values (Henderson et al., 2007). The LUR model estimated concentration of the studied air pollutants in the eighteen districts of Hong Kong was summarized and compared.

**3 Results**

**3.1 Air pollutant measurements**

The measurements at the AQMSs show that the annual-average $PM_{10}$ and $PM_{2.5}$ concentration varied between 30.9–45.7 µg/m³ and 18.4–31.2 µg/m³, respectively (Figure S3), which were higher than the air quality guideline (AQG) for $PM_{10}$ (15.0 µg/m³) and $PM_{2.5}$ (5.0 µg/m³) proposed by the World Health Organization (WHO) (WHO, 2021). $PM_{10}$ TC, $PM_{10}$ $NO_3^-$, $PM_{10}$ $SO_4^{2-}$, and $PM_{10}$ Cd had annual-average concentration of 4511–9019 ng/m³, 2920–4642 ng/m³, 6713–7525 ng/m³, and

0.58–0.72 ng/m$^3$, respectively. The annual-average $NO_2$ concentration was from 9.70 µg/m$^3$ to 197.0 µg/m$^3$, which were generally higher than the WHO $NO_2$ AQG of 10 µg/m$^3$ (WHO, 2021). The annual-average $O_3$ concentration ranged from 17.8 µg/m$^3$ to 73.9 µg/m$^3$ in Hong Kong (Figure S3).

## 3.2 Multi-air-pollutant exposure models

The established annual-average LUR models for ambient $PM_{10}$, $PM_{2.5}$, $NO_2$, $O_3$, and four major $PM_{10}$ chemical species are shown in Table S3. The training $R^2$ values ranged between 0.91 and 0.97, whereas the LOOCV $R^2$ values ranged between 0.73 and 0.93. The results proved that the established LUR models overall achieved relatively good predictive accuracy (Table S3, Figures 2, S4, S5, S6, and S7). The prediction error fractions of the LUR models ranged between -5.9%–7.0%, -6.1%–14%, -4.5%–7.3%, -1.1%–1.2%, -2.3%–3.8%, -8.1%–8.6%, -24%–25%, and -13%–27%, respectively, for $PM_{10}$, $PM_{10}$

TC, $PM_{10}$ $NO_3^-$, $PM_{10}$ $SO_4^{2-}$, $PM_{10}$ Cd, $PM_{2.5}$, $NO_2$, and $O_3$ (Figure 2). There were two to five predictor variables included in the final models. This number of predictor variables was typically within the range of the number of predictor variables in previous studies (Cai et al., 2020; Henderson et al., 2007; Li et al., 2022; Meng et al., 2016; Miri et al., 2019). The selection of these predictor variables was driven by the emission sources of the air pollutants, the dispersion and transport condition, and the influence of transboundary air pollution in Hong Kong. The selected predictor variables included traffic emission-

related variables, different land-use types (e.g., the industrial land), population density, urban morphology (e.g., the canyon height), and geographical locations (Table S3).

Five predictor variables were entered into the $PM_{10}$ LUR model, including the number of private cars in a buffer size of 100 m, the area of buildings within a 100-m buffer, the area of residential land in a buffer size of 100 m, the area of industrial land within a 3000-m buffer, and the area of urban green space in a buffer size of 4000 m. Among these predictor variables,

only the urban green space had a negative direction of effect. The $R^2$ and LOOCV $R^2$ values were 0.92 and 0.77, respectively, representing a remarkable $PM_{10}$ concentration prediction (Table S3). Residual spatial autocorrelation analysis of the $PM_{10}$ LUR model is shown in Figure S8. The $z$-score was 0.399, which means that the model residuals were not spatially correlated, confirming that the $PM_{10}$ LUR model was reasonably established.

For the LUR models of $PM_{10}$ chemical species, two to three predictor variables were selected finally. As expected, the

traffic-related predictor variables and urban/building morphology-related parameters dominated the variability of the four chemical species. It was because the $PM_{10}$ chemical species in the city were mainly influenced by vehicular emissions and urban form patterns (Hsu et al., 2018; Li et al., 2022). For example, the Cd LUR model included the area of transportation land in a buffer size of 2000 m, latitude, and the average canyon height within a 300-m buffer. The $R^2$ ranged from 0.92 to 0.97, whereas the LOOCV $R^2$ values ranged between 0.73 and 0.92, suggesting a relatively high predictive accuracy (Table

S3). The assumption of spatial error independence was confirmed with $z$-score values at -0.249, 0.453, -0.504, and -0.843 for $PM_{10}$ TC, $PM_{10}$ $NO_3^-$, $PM_{10}$ $SO_4^{2-}$, and $PM_{10}$ Cd, respectively (Figure S8).

The PM$_{2.5}$ LUR model included five predictor variables, namely the number of light-duty vehicles in a buffer size of 500 m, the area of urban green space within a 4000-m buffer, the area of residential land in a buffer size of 300 m, the area of buildings within a 50-m buffer, and the maximum building height in a buffer size of 100 m. Four of these variables were the

same as those in the PM$_{10}$ LUR model, even though the buffer sizes varied. The $R^2$ and LOOCV $R^2$ values and the $z$-score values all confirmed that the PM$_{2.5}$ LUR model was reasonably established with an acceptable statistical performance (Table S3 and Figure S8).

The predictor variables included in the NO$_2$ LUR model were the number of total vehicles within a 500-m buffer, the number of people in a buffer size of 100 m, and the area of industrial land within a 1000-m buffer. These predictor variables all had a

positive effect on NO$_2$ concentration, as evidenced by the positive regression slope values. The $R^2$ and LOOCV $R^2$ values were 0.96 and 0.93, respectively, indicating the relatively good prediction performance of the model (Table S3). The model residuals were spatially independent, with a $z$-score value of 0.935 (Figure S8).

The predictor variables included in the O$_3$ LUR model were the number of total vehicles in a buffer size of 700 m, longitude, and the area of urban green space within a 300-m buffer. The predictor variable of vehicles had a negative effect on O$_3$

concentration. This negative effect reflected the titration of O$_3$ in urban areas with a large amount of NO and NO$_2$ emitted by traffic (Han et al. 2023). Longitude had a positive effect on O$_3$ concentration, suggesting the influence of regional transported air masses. A LUR study in Nanjing, China also included longitude in the final O$_3$ model (Huang et al., 2017). Urban green space had a positive effect on O$_3$ concentration, which was probably due to biogenic volatile organic compounds as the precursors of ozone formation (Ma et al., 2021; Ren et al., 2017). The $R^2$ and LOOCV $R^2$ values were 0.92

and 0.87, respectively, showing the relatively high predictive accuracy of the model (Table S3). The $z$-score (1.186) of the residual spatial autocorrelation analysis indicates that the O$_3$ LUR model was well explained by the included predictor variables, with spatially independent model residuals (Figure S8).

### 3.3 Spatial distribution maps

The spatial distribution maps of multiple air pollutants derived from established LUR models are shown in Figure 3, whereas

Table S4 shows the statistical description of the estimated air pollutant concentration. The PM$_{10}$, PM$_{2.5}$, and NO$_2$ LUR models included several predictor variables representing vehicular emissions e.g., the number of medium and heavy-duty vehicles in a buffer size of 500 m (Table S3). Thus, the concentration of PM$_{10}$, PM$_{2.5}$, and NO$_2$ was largely affected by the traffic emissions in Hong Kong, with higher concentration estimated along the road network. In addition, the relatively higher concentration of PM$_{10}$, PM$_{2.5}$, and NO$_2$ was estimated in areas with the high population density (e.g., the northern part

of Hong Kong Island, the Kowloon City district, and the Yau Tsim Mong district). PM$_{10}$ and PM$_{2.5}$ had moderate positive corrections with NO$_2$, with Pearson correlation coefficient (PCC) values of 0.570 and 0.696, respectively (Table 1). Consistent with Li et al. (2022), the LUR model-derived concentration of PM$_{10}$ TC, PM$_{10}$ NO$_3^-$, and PM$_{10}$ SO$_4^{2-}$ was

relatively higher at developed urban areas and along major roads. In contrast to this, the spatial distribution of $PM_{10}$ Cd showed a north–south gradient, with relatively higher concentration in the northern part and relatively lower concentration in the southern part. These $PM_{10}$ chemical species only had weak to moderate positive correlations with $PM_{10}$ mass, with PCC values ranging from 0.189 to 0.589 (Table 1). For $O_3$, there was an increasing trend from west to east, suggesting the influence of transboundary pollution on the spatial distribution pattern. In addition, $O_3$ concentration was largely affected by traffic emissions, with lower concentration estimated along major roads compared with other areas. Due to nitric oxide titration (Han et al., 2023), $O_3$ concentration was generally negatively correlated by various degree with $PM_{10}$, $PM_{10}$ chemical species, $PM_{2.5}$, and $NO_2$ (Table 1).

The spatial patterns of the studied air pollutants varied largely among the districts (Table S5 and Figure S9). As shown by the three example districts in Figure 4, the Yuen Long district had relatively high concentration of $PM_{10}$, $PM_{10}$ species, and $PM_{2.5}$, and moderate concentration of $NO_2$ and $O_3$. For the Yau Tsim Mong district, the $PM_{10}$, $PM_{10}$ species, $PM_{2.5}$, and $NO_2$ concentration was relatively high, whereas the $O_3$ concentration was relatively low. In contrast to this, the Sai Kung district had quite high concentration of $O_3$ but relatively low concentration of other studied air pollutants.

## 4 Discussion and implications

High-density cities usually have spare air quality monitoring stations. This discrepancy clearly highlighted the need to develop LUR models for the spatial mapping of air pollution in high-density cities. This work developed an integrated model framework for assessing multi-air-pollutant exposures in a high-density city based on the air quality data collected at the sparse monitoring stations. Following the proposed integrated model framework, we established multi-air-pollutant exposure models for four major $PM_{10}$ chemical species as well as four criteria gaseous air pollutants in Hong Kong using the LUR model approach (Table S3). Similar to other LUR model studies, one limitation of this study is typically the limited number of monitoring stations. It should be noted that the adequacy of monitoring should not be determined by number of stations alone. This study performed detailed evaluations to examine the adequacy of number of stations. In the GAS module, the established $NO_2$ and $O_3$ exposure models had $R^2$ values of 0.96 and 0.92, respectively, which were similar with previous studies. In the PM module, our $PM_{10}$ and $PM_{2.5}$ exposure models achieved remarkable predictive accuracy, comparable with or higher than those of the traditional LUR studies (Tables 2 and S3; Supplementary Text S2). Following our previous study of Li et al. (2021), in the PM module, we established LUR exposure models of $PM_{10}$ TC, $PM_{10}$ $NO_3^-$, $PM_{10}$ $SO_4^{2-}$, and $PM_{10}$ Cd, with model $R^2$ values higher than 0.92 (Table S3). The detailed evaluation results have proved that our models had promising performance and are capable of reflecting the air quality characteristics of the city. Therefore, our models are considered as sufficient for the scope of this study. Certainty, it is strongly recommended to carry out a further study using different modeling methods (e.g., machine learning) when more data are collected from a larger number of monitoring stations and at a finer temporal resolution.

This research work aimed to contribute to the research area of exposure assessment through providing new opportunities to distinguish the independent associations between combined exposures to multiple air pollutants (i.e., $PM_{10}$, $PM_{10}$ TC, $PM_{10}$ $NO_3^-$, $PM_{10}$ $SO_4^{2-}$, $PM_{10}$ Cd, $PM_{2.5}$, $NO_2$, and $O_3$) and chronic health effects. For example, the finding of weak to moderate spatial correlation between $PM_{10}$ and its chemical species may enable epidemiological studies to separate the chronic health effects of $PM_{10}$ chemical species from the total mass. In addition, the spatial variation of air pollution, together with the geospatial locations of the subjects, can be used for hotspot identification in air quality management and exposure assessment in epidemiological studies (Crouse et al., 2015; Jones et al., 2020; Li et al., 2021). The major explanation for the spatial differences in concentration of multiple air pollutants was the differences in their emission sources (Cai et al., 2020; Jin et al., 2019; Levy et al., 2014; Wu et al., 2017). For instance, $PM_{2.5}$ and $NO_2$ are more linked to traffic and industrial emissions in developed urban areas, while relatively high $O_3$ concentration in rural areas is formed through complex chemical reactions between biogenic volatile organic compounds and nitrogen oxides (Table S3 and Figure 3). The results highlight the importance of the synergistic control of multiple air pollutants and emission sources (Saha et al., 2020; Yim et al., 2019). For instance, the Hong Kong government has spent tremendous efforts on the reduction of vehicular emissions over the past two decades, which successfully reduced traffic-related air pollutants like $PM_{2.5}$ and $NO_2$. However, as revealed by the present study and previous studies (HKEPD, 2022; Zeng et al., 2022), $O_3$ pollution has become an emerging issue, especially in rural areas of Hong Kong. The relationship between the control of vehicular emissions and $O_3$ pollution is complex (Song et al., 2023; Zeng et al., 2022). In Hong Kong, $NO_x$ reductions from the control of vehicular emissions may lead to an increase in the levels of oxidants, and then cause a net $O_3$ production. It is suggested that the control of volatile organic compounds should be implemented to better mitigate $O_3$ pollution in HK (Zeng et al., 2022). This highlights the importance to simulate multiple air pollutants together during exposure assessments. In addition, more research should be conducted to understand the complex and varying interaction of emission sources, pollutant sensitivity to its precursors, and air quality in a city to formulate more effective and specific air quality management policies.

The development and applicability of exposure models depend on the focus of the air pollution epidemiological studies, which either focus on long-term or short-term or even acute exposure. To the best of our knowledge, the annual or long-term LUR exposure models have been widely adopted in providing long-term exposure estimates for health studies (Chen et al., 2021; Wang et al., 2020b). Meanwhile, considering the requirement for high spatial-resolution and short-term acute exposure assessment, it is recommended that more studies should be conducted to establish high spatiotemporal-resolution exposure models when detailed measurement data are available. Indeed, several recent studies have explored the possibility of estimating high spatiotemporal-resolution air pollution exposure models using the spatiotemporal statistical modeling approach (Lu et al., 2020; Masiol et al., 2018). In addition, in future studies, a multi-dimensional and multi-air-pollutant exposure modeling approach is recommended, for instance, by combining spatiotemporal statistical modeling with atmospheric chemistry knowledge, with the particulate chemical species and their toxicity and volatile organic compounds being modeled (Brokamp et al., 2017; Chen et al., 2020b; Lu et al., 2020). In addition, the vertical distribution of air

pollutants should also be measured and modeled to combine with spatiotemporal exposure models to reveal the vertical variability of population exposure to air pollution (Eeftens et al., 2019; Ho et al., 2015; Jin et al., 2019). Moreover, most previous air pollution exposure assessment studies and the current work have focused on ambient air quality, but it is

320 strongly recommended that more research efforts should be made toward developing prediction models of air pollutant concentration in indoor environments (e.g., residential households) for more accurate exposure estimates (Tang et al., 2018).

## 5 Conclusions

In the present study, we developed an integrated model framework for accurate multi-air-pollutant exposure assessments in high-density and high-rise cities. Following the proposed integrated model framework, with the air pollutant concentration

data from a routine monitoring network, annual-average multi-air-pollutant exposure models for ambient $PM_{10}$, major $PM_{10}$ chemical species, $PM_{2.5}$, $NO_2$, and $O_3$ were developed with relatively high predictive performance in Hong Kong, a typical high-rise high-density Asian city. The estimated air pollution maps (500 m $\times$ 500 m resolution) of these air pollutant mixtures could be used to support a unique combined exposure assessment in health studies.

We anticipate that the proposed integrated model framework can be easily extended to establish multi-air-pollutant exposure

models in other cities. Apart from the LUR approach, other spatiotemporal statistical modeling methods, such as various machine learning algorithms, should be applied when a larger data set is available. Particularly, the development of high spatiotemporal exposure models should be explored when a high temporal-resolution air pollutant measurement data set is collected. Furthermore, the associations of combined exposures to multiple air pollutants with health endpoints should be analysed to provide new insights on the health-oriented air pollution control.

*Data availability.* The model data presented in this article are available from the authors upon request (yimsteve@gmail.com).

*Author contributions.* ZYL and SHLY designed the study. ZYL performed all the data analysis with support from SHLY. ZYL wrote the paper with contributions from all co-authors.

*Competing interests.* The authors declare that they have no conflict of interest.

*Acknowledgements.* We would like to thank the Hong Kong Environmental Protection Department and the Hong Kong Observatory for providing air quality and meteorological data, respectively.

*Financial support.* This work is funded by the Vice-Chancellor's Discretionary Fund of The Chinese University of Hong Kong (grant no. 4930744), the Dr. Stanley Ho Medical Development Foundation (grant no. 8305509), and the project from the ENvironmental SUstainability and REsilience (ENSURE) partnership between the CUHK and UoE. ZYL was supported by the "100-top-talents Program" Start-up Grant of Sun Yat-sen University (Grant No. 220204).

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

Table 1. Pearson correlation coefficients (PCCs) among the LUR estimated concentration of ambient $PM_{10}$, $PM_{10}$ TC, $PM_{10}$ $NO_3^-$, $PM_{10}$ $SO_4^{2-}$, $PM_{10}$ Cd, $PM_{2.5}$, $NO_2$, and $O_3$ in Hong Kong.

| | $PM_{10}$ | $PM_{10}$ TC | $PM_{10}$ $NO_3^-$ | $PM_{10}$ $SO_4^{2-}$ | $PM_{10}$ Cd | $PM_{2.5}$ | $NO_2$ | $O_3$ |
|---|---|---|---|---|---|---|---|---|
| $PM_{10}$ | 1 | 0.397** | 0.589** | 0.189** | 0.430** | 0.781** | 0.570** | -0.354** |
| $PM_{10}$ TC | | 1 | 0.747** | 0.418** | 0.432** | 0.598** | 0.785** | -0.677** |
| $PM_{10}$ $NO_3^-$ | | | 1 | 0.495** | 0.479** | 0.729** | 0.921** | -0.654** |
| $PM_{10}$ $SO_4^{2-}$ | | | | 1 | 0.192** | 0.256** | 0.488** | -0.383** |
| $PM_{10}$ Cd | | | | | 1 | 0.368** | 0.522** | -0.174** |
| $PM_{2.5}$ | | | | | | 1 | 0.696** | -0.500** |
| $NO_2$ | | | | | | | 1 | -0.678** |
| $O_3$ | | | | | | | | 1 |

** Correlation is significant at the 0.01 level (2-tailed).

Table 2. A comparison of this study with previous LUR studies. The example LUR studies focusing on at least one of the criteria air pollutants published in recent five years were included.

| Study area | $PM_{10}$ | $PM_{10}$ TC | $PM_{10}$ $NO_3^-$ | $PM_{10}$ $SO_4^{2-}$ | $PM_{10}$ Cd | $PM_{2.5}$ | $NO_2$ | $O_3$ | References |
|---|---|---|---|---|---|---|---|---|---|
| Hong Kong, China | 0.92 | 0.94 | 0.93 | 0.97 | 0.92 | 0.91 | 0.96 | 0.92 | This work |
| Beijing, China | | | | | | 0.86 | | | Xu et al. (2019) |
| Tianjin, China | | | | | | | | 0.98 | Wang et al. (2020) |
| Nanjing, China | | | | | | 0.75 | 0.87 | 0.65 | Huang et al. (2017) |
| Lanzhou, China | | | | | | 0.77 | 0.71 | | Jin et al. (2019) |
| Hong Kong, China | | | | | | 0.59 | 0.46 | | Lee et al. (2017) |
| Southern California, USA | | | | | | 0.47 | | | Jones et al. (2020) |
| Sabzevar, Iran | 0.75 | | | | | 0.71 | | | Miri et al. (2019) |
| Auckland, New Zealand | | | | | | | 0.66 | | Ma et al. (2019) |
| Mexico City, Mexico | 0.73 | | | | | 0.83 | 0.81 | | Son et al. (2018) |
| Augsburg, Germany | 0.91 | | | | | 0.84 | 0.95 | 0.92 | Wolf et al. (2017) |
| Manchester, UK | 0.95 | | | | | | | | Mölter and Lindley (2021) |
| Sydney, Australia | | | | | | | 0.84 | | Cowie et al. (2017) |


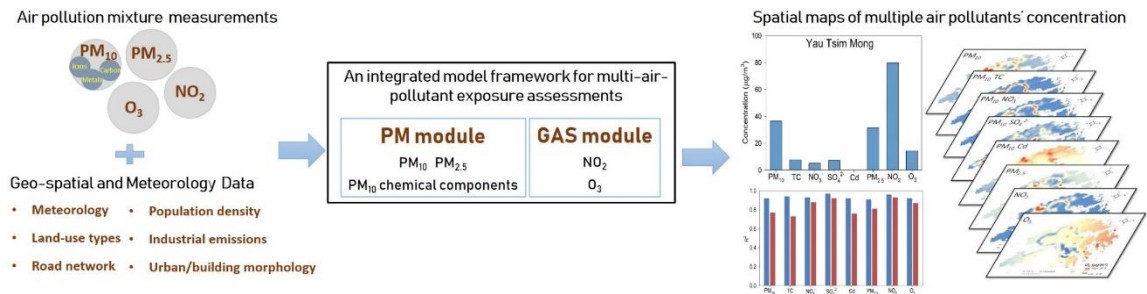

Figure 1. An integrated model framework for multi-air-pollutant exposure assessments in high-density cities. It mainly includes two components of particulate matters (PM module) and gaseous pollutants (GAS module). PM module consists of the measurement and LUR modeling of $PM_{2.5}$, $PM_{10}$, and $PM_{10}$ major chemical components (e.g., ions, metals, and carbon), while GAS module involves the measurement and LUR modeling of $NO_2$ and $O_3$.

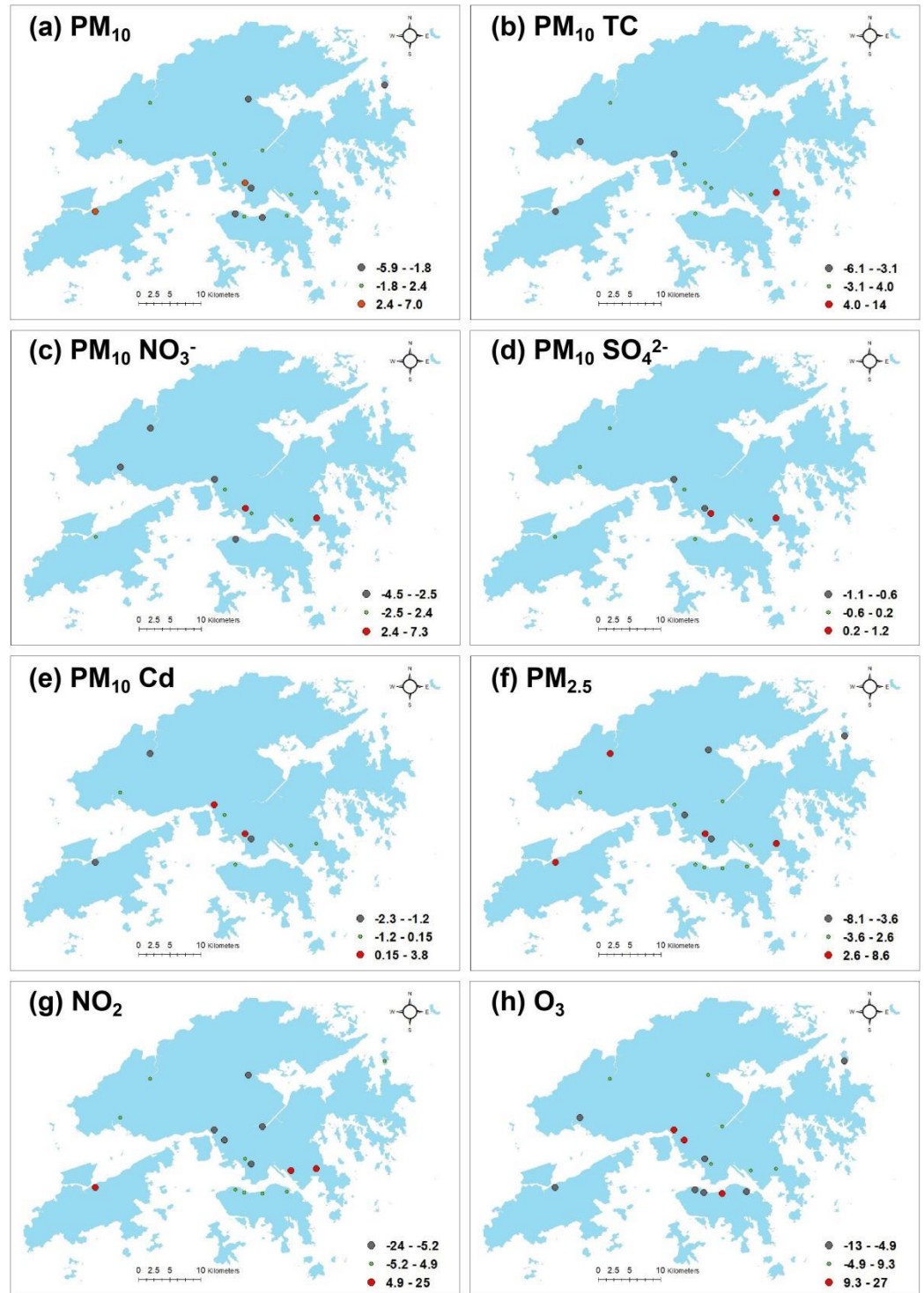

Figure 2. The distribution of prediction error fractions (%) of the established LUR models. The prediction error fraction is defined as [(predicted concentration – observed concentration)/observed concentration].

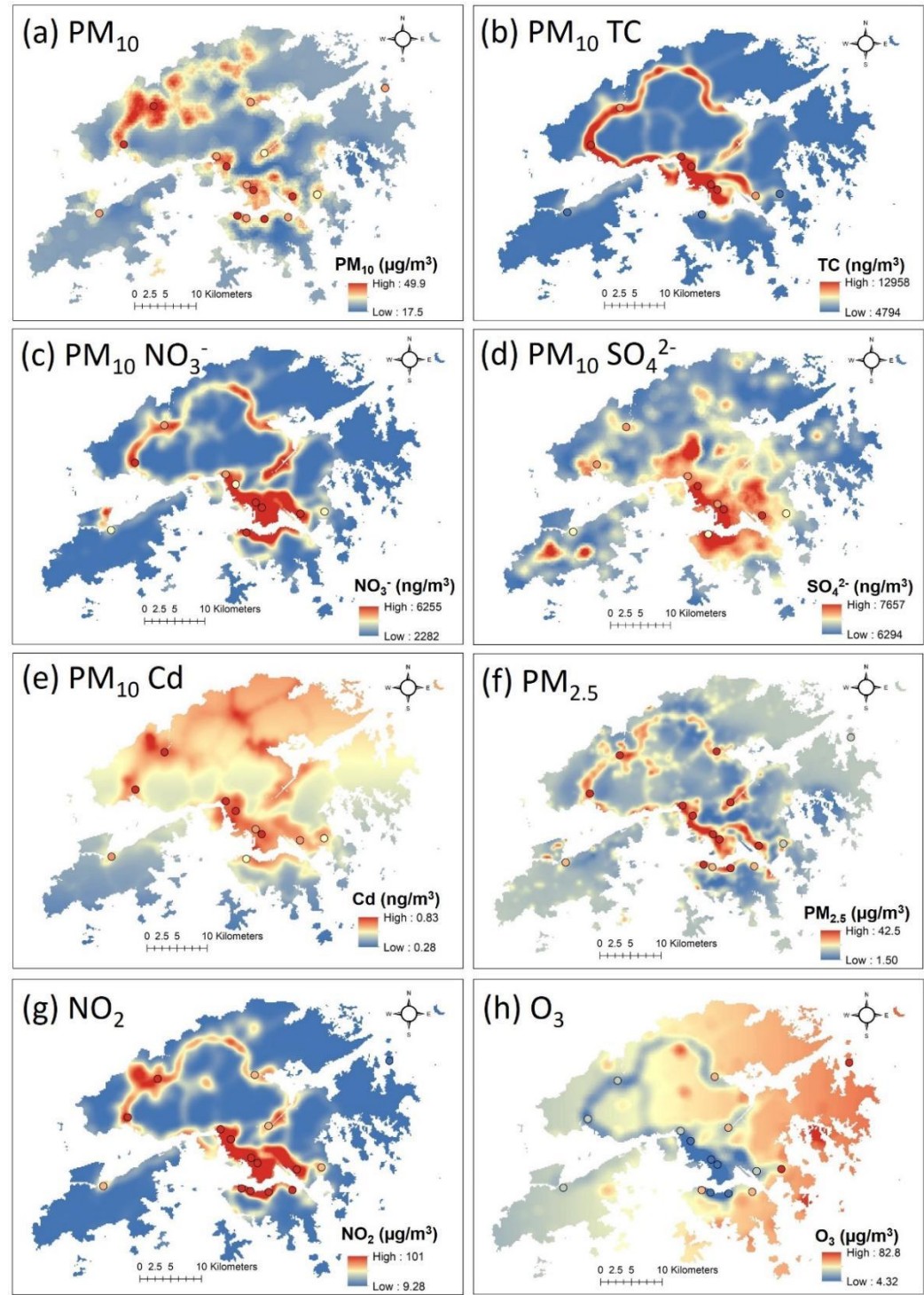

Figure 3. LUR model-derived spatial distribution maps of annual-average ambient $PM_{10}$, $PM_{10}$ TC, $PM_{10}$ $NO_3^-$, $PM_{10}$ $SO_4^{2-}$, $PM_{10}$ Cd, $PM_{2.5}$, $NO_2$, and $O_3$ concentration in Hong Kong. The colored circles represent observations at AQMSs.

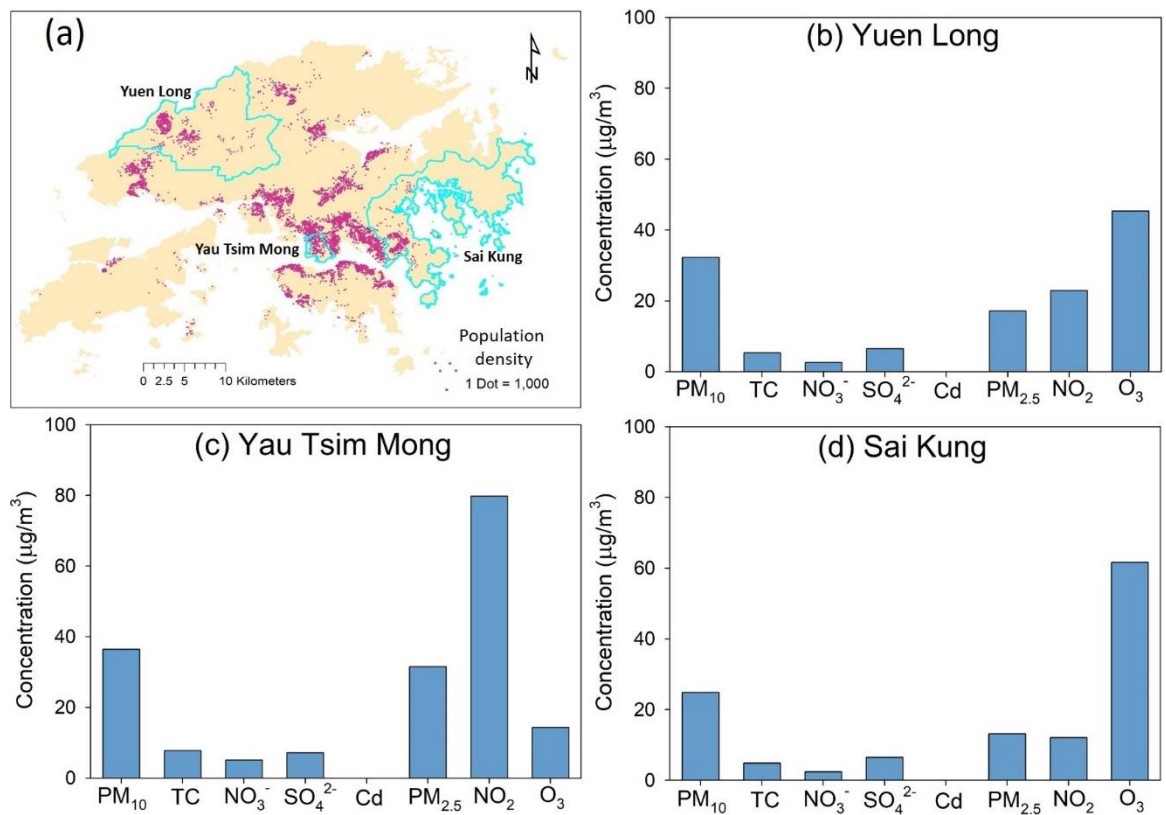

Figure 4. The average LUR estimated ambient $PM_{10}$, $PM_{10}$ TC, $PM_{10}$ $NO_3^-$, $PM_{10}$ $SO_4^{2-}$, $PM_{10}$ Cd, $PM_{2.5}$, $NO_2$, and $O_3$ concentration in three representative districts in Hong Kong. (a) The distribution of Yuen Long, Yau Tsim Mong, and Sai Kung districts in Hong Kong with the population density shown. (b) Yuen Long (a district more influenced by the transboundary pollution). (c) Yau Tsim Mong (a high-density district more influenced by vehicular emissions). (d) Sai Kung (a rural district).