# Peer review of "Development of an integrated model framework for multi-air-pollutant exposure assessments in high-density cities"

_EGUsphere, 2023_

## Author Comment (AC1)

Dear referee,

Thank you for your comments and suggestions. We carefully addressed them one by one as shown below. Hope you find our revisions useful. Thank you again.

Regards,

YIM, Hung-Lam Steve, Ph.D.

Associate Professor, Asian School of the Environment
Associate Professor, Lee Kong Chian School of Medicine
Principal Investigator, Earth Observatory of Singapore
Nanyang Technological University (NTU), Singapore

Email: steve.yim@ntu.edu.sg
ASE@NTU: https://www.ntu.edu.sg/ase/aboutus/staff-directory/staff-details/yim-hung-lam-steve
Address: Block N2-01C-44, Asian School of the Environment, Nanyang Technological University, 50 Nanyang Avenue, Singapore, 639798

**#Referee 1**
**The authors present results from a land use regression (LUR) framework used to create 500m resolution exposure fields for multiple pollutants, including NO2, O3, total PM1- and PM2.5, and multiple PM10 species. Their LUR models have good predictive performance in leave one out cross validation. I agree with the authors that the fine-scale exposure products may be useful for future exposure and epidemiological studies. I believe, however, that the study does not go far enough in explaining the implications of their results. In particular, I find it is lacking in two areas relevant to the scope of ACP.**
**First, very little description is given of the meaning behind the variables that are selected for the LUR models, including why they are predictive of the various species and what they may be proxies for. Did any variables that have been found predictive not make sense? Are there co-linearities that may be obscuring the influence of some variables over others? Can we learn something from the predictors chosen that can inform policies to reduce exposure?**

**Response:** The manuscript has provided descriptions to explain the associations between air pollutants and their predictive variables. The descriptions can be found in Line 221-226, Line 231-247, Line 150-151 and Line 273-280. The explanations were confirmed by the supports of previous research. Thus, the meaning behind the relationships were clearly demonstrated.

It is true that some variables may have a certain level of co-linearities. Similar to previous studies, we applied Variance Inflation Factor (VIF) measures the severity of multicollinearity in regression analysis. Predictor variables with a value higher than 5.0 were removed (de Hoogh et al., 2013; Gulliver et al., 2018; Hsu et al., 2018; Jones et al., 2020; Larkin et al., 2017; Ma et al., 2019; Zhang et al., 2015). So, it is believed that co-linearities should not induce major issues.

The model was demonstrated its capabilities to simulate the multiple air pollutants, and to reflect the relationships between each predictive variable and various air pollutants. Similar to other studies (Vizcaino and Lavalle, 2018; Shi et al., 2020), our model can be used to evaluate any potential air pollutant emission control policies especially for those targeting to multiple air pollutants.

The descriptions in the main paper:

*Line 221-226: The predictor variables included in the $O_3$ LUR model were the number of total vehicles in a buffer size of 700 m, longitude, and the area of urban green space within a 300-m buffer. The predictor variable of vehicles had a negative effect on $O_3$ concentration. This negative effect reflects the titration of $O_3$ in urban areas with a large amount of NO and $NO_2$ emitted by traffic. Longitude has a positive effect on $O_3$ concentration, suggesting the influence of regional transported air masses. A LUR study in Nanjing, China also included longitude in the final $O_3$ model (Huang et al., 2017). Urban green space had a positive effect on $O_3$ concentration, which is probably due to biogenic volatile organic compounds as the precursors of ozone formation (Ma et al., 2021; Ren et al., 2017).*

*Line 231-247: The $PM_{10}$, $PM_{2.5}$, and $NO_2$ LUR models included several predictor variables (e.g., the number of medium and heavy-duty vehicles in a buffer size of 500 m) representing vehicular emissions (Table S3). Thus, the concentration of $PM_{10}$, $PM_{2.5}$, and $NO_2$ is largely affected by the traffic emissions in Hong Kong, with higher concentration estimated along the road network. In addition, the relatively higher concentration of $PM_{10}$, $PM_{2.5}$, and $NO_2$ was estimated in areas with the high population density (e.g., the northern part of Hong Kong island, the Kowloon City district, and the Yau Tsim Mong district). $PM_{10}$ and $PM_{2.5}$ had moderate positive corrections with $NO_2$, with Pearson correlation coefficient (PCC) values at 0.570 and 0.696, respectively (Table 1). Consistent with Li et al. (2022), the LUR model-derived concentration of $PM_{10}$ TC, $PM_{10}$ $NO_3^-$, and $PM_{10}$ $SO_4^{2-}$ was relatively higher at developed urban areas and along major roads. In contrast to this, the spatial distribution of $PM_{10}$ Cd showed a north–south gradient, with relatively higher concentration in the northern part and relatively lower concentration in the southern part. These $PM_{10}$ chemical species only had weak to moderate positive correlations with $PM_{10}$ mass, with PCC values ranging from 0.189 to 0.589 (Table 1). For $O_3$, there was an increasing trend from west to east, suggesting the influence of transboundary pollution on the spatial distribution pattern. In addition, $O_3$ concentration was largely affected by traffic emissions, with lower concentration estimated along major roads compared with other areas. Due to nitric oxide titration (Han et al., 2023), $O_3$ concentration was generally negatively correlated by various degree with $PM_{10}$, $PM_{10}$ chemical species, $PM_{2.5}$, and $NO_2$ (Table 1).*

*Line 150-151: Due to this procedure, the included predictor variables may obscure the potential influence of others.*

*Line 273-280: The results highlight that the synergistic control of multiple emission sources and key precursors is urgently needed for the joint control of multiple air pollutants (Saha et al., 2020; Yim et al., 2019). For instance, the Hong Kong government has spent tremendous efforts on the reduction of vehicular emissions over the past two decades, which successfully reduced traffic-related air pollutants like $PM_{2.5}$ and $NO_2$. However, as revealed by the present study and previous studies (HKEPD, 2022; Zeng et al., 2022), $O_3$ pollution has become an emerging issue, especially in rural areas of Hong Kong, which cannot be accomplished*

*through the control of vehicular emissions. Thus, more research efforts should be conducted to understand the complex and varying interaction of emission sources, pollutant sensitivity to precursors, and air quality in a city to formulate more effective and specific air quality management policies.*

**Second, there is little description provided of the exposure products themselves. What are the implications for human exposure? It would be useful to select important areas of the city (e.g., an area with high population density), describe which pollutants are predicted to have high concentrations, and offer some suggestions about why.**

**Response:** This manuscript mainly focused on the development of an integrated model framework. Due to the length limit, it is not feasible to provide detailed description of the exposure products. Nevertheless, we did try our best to describe the exposure results. Section 3.3 provided spatial distribution map in section 3.3 , while the description in Line 267-280 highlighted the mechanisms and insights for pollution control.

The descriptions in the main paper:

*3.3 Spatial distribution maps: The spatial distribution maps of multiple air pollutants derived from established LUR models are shown in Figure 2, whereas Table S4 shows the statistical description of the estimated air pollutant concentration. The $PM_{10}$, $PM_{2.5}$, and $NO_2$ LUR…*

*Line 267-280: The spatial variation of air pollution can be used for hotspot identification for air quality management and exposure assessment in epidemiological studies using the geospatial locations of the subjects (Crouse et al., 2015; Jones et al., 2020; Li et al., 2021). The major explanation for the spatial differences in concentration of multiple air pollutants was the differences in their emission sources (Cai et al., 2020; Jin et al., 2019; Levy et al., 2014; Wu et al., 2017). For instance, $PM_{2.5}$ and $NO_2$ are more linked to traffic and industrial emissions in developed urban areas, while relatively high $O_3$ concentration in rural areas is formed through complex chemical reactions between biogenic volatile organic compounds and nitrogen oxides (Table S3 and Figure 2). The results highlight that the synergistic control of multiple emission sources and key precursors is urgently needed for the joint control of multiple air pollutants (Saha et al., 2020; Yim et al., 2019). For instance, the Hong Kong government has spent tremendous efforts on the reduction of vehicular emissions over the past two decades, which successfully reduced traffic-related air pollutants like $PM_{2.5}$ and $NO_2$. However, as revealed by the present study and previous studies (HKEPD, 2022; Zeng et al., 2022), $O_3$ pollution has become an emerging issue, especially in rural areas of Hong Kong, which cannot be accomplished through the control of vehicular emissions. Thus, more research efforts should be conducted to understand the complex and varying interaction of emission sources, pollutant sensitivity to precursors, and air quality in a city to formulate more effective and specific air quality management policies.*

**Specific comments**
**Abstract: it is important to describe the exposure product in full, including the temporal coverage (i.e., which year?)**

**Response:** Revised as suggested (Line 20-21).
Line 20-21: …2017 annual-average exposures of four major $PM_{10}$ chemical species as well as four criteria air pollutants of $PM_{10}$, $PM_{2.5}$, $NO_2$, and $O_3$ in…

**Line 81-82: it strikes me as strange to have 3 citations for a sentence describing the geography of Hong Kong.**

**Response:** Revised with only one reference kept (Line 85).
Line 85: …the southeast coast of the Pearl River Delta (PRD) region, China (Yim et al., 2009).

**122-123: it would be helpful to explain more about why lat/lon would account appropriately for transboundary pollution**

**Response:** We added the capacity of using lat/lon as predictor variables in the revised manuscript (Line 128-129).

*Line 128-129: …, the geo-locations of longitude and latitude were also adopted which could reveal a north-south or west-east gradient of air pollutant concentrations (Huang et al., 2017).*

**134-135: please explain "if the direction was as pre-defined"**

**Response:** We revised the text to improve this description (Line 142).

*Line 142: …the direction was consistent with the pre-defined one.*

**135: model selection process: please confirm whether the model selection R2 was calculated on the training dataset or the hold out dataset. If the training dataset, is there risk of over-fitting?**

**Response:** The calculation was on the full dataset (Line 140). We then further used leave-one-out cross-validation (LOOCV) method to validate the established model (Line 158).

*Line 140: Using the full dataset, we ranked…*
*Line 158: Leave-one-out cross-validation (LOOCV) was used…*

**135: It seems to me that more flexible machine learning methods may be more adept at capturing nonlinear relationships between environmental predictors and measured pollutants. Is it expected that these variables have a linear relationship with pollutant species? The final paragraph of the manuscript mentions that these are more appropriate with more data, but there is no evidence given or description of how much data is needed.**

**Response:** It is correct that environmental predictors and measured air pollutants may have nonlinear relationships. Nevertheless, LUR models are still the widely used model for epidemiological research because they are well proved to reflect the relationships between predictive variables and targeted air pollutants. There are still a large number of air quality modelling studies using LUR models (de Hoogh et al., 2013; Gulliver et al., 2018; Hsu et al., 2018; Jones et al., 2020; Larkin et al., 2017; Ma et al., 2019; Zhang et al., 2015; Vizcaino and Lavalle, 2018; Shi et al., 2020). We agree that machine learning method is another a useful tool for air quality modelling, so our main paper has included the further research perspective on using various machine learning algorithms (Line 305-306).

Regarding the required data set for LUR/machine learning model development, it is an important research topic but out of the scope of the present study, which mainly proposes an integrated model framework for accurate multi-air-pollutant exposure assessments in high-density and high-rise cities.

*Line 305-306: Apart from the LUR approach, other spatiotemporal statistical modelling methods, such as various machine learning algorithms, should be applied when a larger data set is available.*

**167: "either comparable to or higher than" makes it sound like a dichotomous variable. I think "generally greater than" would describe this well enough.**

**Response:** Revised as suggested (Line 182).

*Line 182: ... generally higher than...*

**176-178: I do not understand how the selection of the predictor variables was related to these other factors. Please clarify.**

**Response:** We revised the mentioned content to make it clearer (Line 191-192).

*Line 191-192: The selection of these predictor variables was driven by the emission sources...*

**192: This line refers to a negative z score close to 1. Does this suggest anti-spatial correlation for Cd? More description of this variable is needed**

**Response:** We added the description of Moran's $I$ and the corresponding $z$-score and $p$-values in the revised manuscript (Line 152-156).

*Line 152-156: The spatial autocorrelation (Moran's I) tool measures spatial autocorrelation using both feature locations and feature values simultaneously. Meanwhile, z-score and P-values were calculated to evaluate the significance of Moran's I value. z-score values are standard deviations, whereas the Moran's I index is bounded by -1.0 and 1.0. When the z-score or P-value indicates statistical significance, a positive Moran's I index value indicates tendency towards clustering, whereas a negative Moran's I index value indicates tendency towards dispersion (Cordioli et al., 2017; Luminati et al., 2021). Moran's I index and the corresponding z-score and P values on...*

**Could the authors provide spatial maps of error at monitor locations? In general, it would be useful to develop estimates of uncertainty on the the same spatial scale as the predictions.**

**Response:** We have added the spatial maps of prediction error [differences between predictions and observations] (Figure S7 in SI) and error fraction [(predictions-observations)/observations] (Figure 2) of the LUR models to show the uncertainty/error. The related content is added (Line 186-189).

[Figure]

Figure S8. The distribution of prediction errors [predicted concentration – observed concentration] of the established LUR models.

[Figure]

Figure 2. The distribution of prediction error fractions (%) of the established LUR models. The prediction error fraction is defined as [(predicted concentration – observed concentration)/observed concentration].

*Line 186-189: The prediction error fractions of the LUR models ranged between -5.9%–7.0%, -6.1%–14%, -4.5%–7.3%, -1.1%–1.2%, -2.3%–3.8%, -8.1%–8.6%, -24%–25%, and -13%–27%, respectively, for $PM_{10}$, $PM_{10}$ TC, $PM_{10}$ $NO_3^-$, $PM_{10}$ $SO_4^{2-}$, $PM_{10}$ Cd, $PM_{2.5}$, $NO_2$, and $O_3$ (Figure 2).*

**220: is this implying that population density leads to high PM? Can the authors suggest a mechanism here that isn't explained by the other model covariates?**

**Response:** Population density can serve as a proxy to reflect the contribution of other emission sources due to human activities (i.e. cooking and industrial emissions) that were not included in the model. We do include the explanation on this mechanism (Line 269-271).

*Line 269-271: The major explanation for the spatial differences in concentration of multiple air pollutants was the differences in their emission sources (Cai et al., 2020; Jin et al., 2019; Levy et al., 2014; Wu et al., 2017). For instance, $PM_{2.5}$ and $NO_2$ are more linked to traffic and industrial emissions in developed urban areas, ...*

**239-241: I am confused about the difference between the GAS and PM modules. Is it necessary to differentiate beyond the pollutant species names?**

**Response:** In the revised manuscript, we added the reason for separating PM and GAS modules (Line 76-79).

*Line 76-79: The PM and GAS modules were separated because the measurement and LUR modelling of PM species and gaseous pollutants are largely different in terms of measurement techniques, the number of required measurement sites, and selected predictor variables, etc. the In the present study, the...*

**257-260: I am not sure what this adds to the discussion. Can the authors be more specific here based on the predictors chosen for the final model?**

**Response:** This part aimed at providing a description to link our established models to policy evaluation. We have revised the mentioned content to make it more specific (Line 274-280).

*Line 274-280: For instance, the Hong Kong government has spent tremendous efforts on the reduction of vehicular emissions over the past two decades, which successfully reduced traffic-related air pollutants like $PM_{2.5}$ and $NO_2$. However, as revealed by the present study and previous studies (HKEPD, 2022; Zeng et al., 2022), $O_3$ pollution has become an emerging issue, especially in rural areas of Hong Kong, which cannot be accomplished through the control of vehicular emissions. Thus, more research efforts should be conducted to understand the complex and varying interaction of emission sources, pollutant sensitivity to precursors, and air quality in a city to formulate more effective and specific air quality management policies.*

**How was 500m selected as the best resolution for the predictions?**

**Response:** Based on the spatial resolution of the predictor variables. We revised the related content to make it clearer (Line 169-170).

*Line 169-170: A spatial resolution of 500 m ×500 m was adopted here due to the spatial resolution of most predictor variables is at several hundred meters resolution.*

**Reference:**

Declercq, C., Dėdelė, A., Dons, E., de Nazelle, A., Eeftens, M., Eriksen, K., Eriksson, C., Fischer, P., Gražulevičienė, R., Gryparis, A., Hoffmann, B., Jerrett, M., Katsouyanni, K., Iakovides, M., Lanki, T., Lindley, S., Madsen, C., Mölter, A., Mosler, G., Nádor, G., Nieuwenhuijsen, M., Pershagen, G., Peters, A., Phuleria, H., Probst-Hensch, N., Raaschou-Nielsen, O., Quass, U., Ranzi, A., Stephanou, E., Sugiri, D., Schwarze, P., Tsai, M.-Y., Yli-Tuomi, T., Varró, M. J., Vienneau, D., Weinmayr, G., Brunekreef, B., and Hoek, G.: Development of Land Use Regression Models for Particle Composition in Twenty Study Areas in Europe, Environ. Sci. Technol., 47, 5778–5786, https://doi.org/10.1021/es400156t, 2013.

Gulliver, J., Morley, D., Dunster, C., McCrea, A., van Nunen, E., Tsai, M.Y., Probst-Hensch, N., Eeftens, M., Imboden, M., Ducret-Stich, R., and Naccarati, A.: Land use regression models for the oxidative potential of fine particles (PM2.5) in five European areas. Environ. Res. 160, 247-255, 2018.

Hsu, C.Y., Wu, C.D., Hsiao, Y.P., Chen, Y.C., Chen, M.J., and Lung, S.C.C.: Developing land-use regression models to estimate PM2.5-bound compound concentrations. Remote Sens. 10(12), 1971, 2018.

Jones, R.R., Hoek, G., Fisher, J.A., Hasheminassab, S., Wang, D., Ward, M.H., Sioutas, C., Vermeulen, R., and Silverman, D.T.: Land use regression models for ultrafine particles, fine particles, and black carbon in southern California. Sci. Total Environ. 699, 134234, 2020.

Larkin, A., Geddes, J. A., Martin, R. V., Xiao, Q., Liu, Y., Marshall, J. D., Brauer, M., and Hystad, P.: Global Land Use Regression Model for Nitrogen Dioxide Air Pollution, Environ. Sci. Technol., 51, 6957–6964, https://doi.org/10.1021/acs.est.7b01148, 2017.

Ma, X., Longley, I., Gao, J., Kachhara, A., and Salmond, J.: A site-optimised multi-scale GIS based land use regression model for simulating local scale patterns in air pollution. Sci. Total Environ. 685, 134-149, 2019.

Shi, Y., Bilal, M., Ho, H. C., and Omar, A.: Urbanization and regional air pollution across South Asian developing countries – A nationwide land use regression for ambient PM2.5 assessment in Pakistan, Environmental Pollution, 266, 115145, https://doi.org/10.1016/j.envpol.2020.115145, 2020.

Vizcaino, P. and Lavalle, C.: Development of European NO2 Land Use Regression Model for present and future exposure assessment: Implications for policy analysis, Environmental Pollution, 240, 140–154, https://doi.org/10.1016/j.envpol.2018.03.075, 2018.

Zhang, J.J., Sun, L., Barrett, O., Bertazzon, S., Underwood, F.E., and Johnson, M.: Development of land-use regression models for metals associated with airborne particulate matter in a North American city. Atmos. Environ. 106, 165-177, 2015.

---

## Author Comment (AC2)

Dear referee,

Thank you for your comments and suggestions. We carefully addressed them one by one as shown below. Hope you find our revisions useful. Thank you again.

Regards,

YIM, Hung-Lam Steve, Ph.D.

Associate Professor, Asian School of the Environment
Associate Professor, Lee Kong Chian School of Medicine
Principal Investigator, Earth Observatory of Singapore
Nanyang Technological University (NTU), Singapore

Email: steve.yim@ntu.edu.sg
ASE@NTU: https://www.ntu.edu.sg/ase/aboutus/staff-directory/staff-details/yim-hung-lam-steve
Address: Block N2-01C-44, Asian School of the Environment, Nanyang Technological University, 50 Nanyang Avenue, Singapore, 639798

**#Referee 2**
**The author employs a land-use regression (LUR) approach in developing an integrated model framework to estimate the annual average exposures of four primary PM10 chemical species and four criteria air pollutants—PM10, PM2.5, NO2, and O3—within the context of a typical high-rise and high-density Asian city, namely Hong Kong, China. The methodology leverages annual concentrations of these air pollutants as captured by monitoring stations in Hong Kong. However, it appears there are only sixteen data points available for analysis, and in certain cases, even less for some pollutants.**
**Here are some significant queries I would like the author to address:**
**The choice of 2017 as the year of interest is based on the completeness of data. Nevertheless, how is this year relevant to the epidemiological data that the author plans to investigate further? The applicability of the annual model to current epidemiological studies seems somewhat limited. Given the study's title, could the author shed more light on this concern?**

**Response:** Our models are particularly useful for cross-sectional health studies. It is because our model can provide multiple air pollutants over space at high spatial resolution that monitoring stations cannot provide. For long-term exposure, multiple models can be established for other years. This study demonstrated the development of an integrated model framework using the data in 2017 that is the year our research team collected health data in a Hong Kong cohort. In the revised manuscript, we added content on the applicability of exposure models (Line 281-288).

*Line 281-288: The development and applicability of exposure models depends on the focus of the air pollution epidemiological studies, which either focus on long-term or short-term or even acute exposure. To the best of our knowledge, the annual or long-term LUR exposure models have been widely adopted in providing long-term exposure estimates for health studies. Meanwhile, considering the requirement for high spatial-resolution and short-term acute*

*exposure assessment, it is recommended that more studies are conducted to establish high spatiotemporal-resolution exposure models when detailed measurement data are available. Indeed, several recent studies have explored the possibility of estimating high spatiotemporal-resolution air pollution exposure models using the spatiotemporal statistical modelling approach (Lu et al., 2020; Masiol et al., 2018).*

**The author refers to the work as an "integrated model framework". However, I can only discern individual LUR models designated for each pollutant. Could the author elaborate on how the integration of this model framework was accomplished?**

**Response:** This work provided an integrated model framework to simulate multiple air pollutants based on one set of input data. It should be noted that the PM and GAS modules that are consisted of multiple LUR models require different type and number of predictive variables. The integrated model framework handles the input data required for the PM and GAS modules and provides the results of eight air pollutants. It is therefore considered an integrated model framework.

In the revised manuscript, we added the description on the integration of the PM and GAS modules (Line 76-82).

*Line 76-82: The PM and GAS modules were separated because the measurement and LUR modelling of PM species and gaseous pollutants are largely different in terms of measurement techniques, the number of required measurement sites, and selected predictor variables, etc. In the present study, the PM module includes the LUR models for different sizes of PM and its chemical components, whereas the GAS module includes the LUR models for two typical gaseous pollutants of $NO_2$ and $O_3$. The included air pollutants vary depending on the data availability when the proposed integrated model framework is applied in other cities.*

**The models were specifically developed for "high-density cities". Still, it appears that there is a mismatch between the sparse density of monitoring stations and the actual population density. Could the author provide further clarification on this discrepancy?**

**Response:** This is an important point. The city has a high population density but monitoring stations are sparse. The discrepancy clearly highlighted the need to develop a LUR model for high-density cities. In addition to number of stations, LUR model development also requires the diversity of monitoring stations. So, this work aimed at developing an integrated model framework based on the data collected at the sparse monitoring stations for high-density cities. In the revised manuscript, we added the general diverse and representative distribution of the included monitoring stations (Line 100-102).

*Line 100-102: These AQMSs are generally diverse and representative, ranging from rural stations under a limited influence of anthropogenic emissions to traffic stations near the major roads in Hong Kong.*

**Overall, it appears that the number of data points presents a significant limitation in this study. Would the author consider adopting a spatio-temporal model to incorporate a larger data set with finer temporal resolution?**

**Response:** Similar to other LUR model studies, one limitation is typically the limited number of stations. To determine whether the limitation is significant or not cannot be based on the

number alone. Instead, it should base on detailed evaluations as we did in this work. We have proved that our models had promising performance and are capable of reflecting the air quality characteristics of the city. Therefore, our models as considered as sufficient for the scope of this paper. Certainly, it would be a good idea for a further study if there are data at a finer temporal resolution. Thank you for your suggestion.

---

## Author Response (AR2)

Dear editor and reviewers,

 Thank you for your comments and suggestions. We carefully addressed them one by one as shown below. Hope you find our revisions useful. Thank you again.

Regards,

YIM, Hung-Lam Steve, Ph.D.

Associate Professor, Asian School of the Environment
Associate Professor, Lee Kong Chian School of Medicine
Principal Investigator, Earth Observatory of Singapore
Nanyang Technological University (NTU), Singapore

Email: steve.yim@ntu.edu.sg
ASE@NTU: https://www.ntu.edu.sg/ase/aboutus/staff-directory/staff-details/yim-hung-lamsteve
Address: Block N2-01C-44, Asian School of the Environment, Nanyang Technological University, 50 Nanyang Avenue, Singapore, 639798

1. **Regarding this comment "The choice of 2017 as the year of interest is based on the completeness of data. Nevertheless, how is this year relevant to the epidemiological data that the author plans to investigate further? The applicability of the annual model to current epidemiological studies seems somewhat limited. Given the study's title, could the author shed more light on this concern?" I feel that the authors gave a good response, but edits made were entirely in the discussion at the end of the paper. If "implications for epidemiological research" is really important as the title would suggest, then consider discussing this context in the Introduction. I would also encourage the authors to consider removing "implications for epidemiological research" from the title, since it is not clear to me that this is a major focus of the paper.**

**Response:** Thanks for the comment. Taking into the reviewer's comment and your suggestion, we decide to remove the "the implication for epidemiological research". Nevertheless, in the *Introduction* section, we still mention the benefit to develop an integrated model framework for supporting epidemiological studies (Line 50-54, Line 55-56, Line 63-67).

2. **Regarding this comment "The author refers to the work as an "integrated model framework". However, I can only discern individual LUR models designated for each pollutant. Could the author elaborate on how the integration of this model framework was accomplished?" The authors responded by clarifying that there are PM and GAS modules. But the reviewer's question remains unanswered from what I can tell. In what way is this framework integrated? Are separate LUR models created for each**

**pollutant independently? Do measurements of one pollutant in any way influence predictions of another pollutant? It seems that more clarity is important to understand this integrated framework. Also, what are the roles of the two modules and their interactions within the framework?**

**Response:** Thanks for the comment. Our model framework is developed to provide a platform allowing users to run the two modules with multiple air pollutants by inputting one set of input data. We selected the term "integration" instead of "coupling" because the results of one module do not affect the calculation of another module. In the revised manuscript, we clarify the model development of PM and GAS modules and their roles and interactions (Line 79-84).

Line 79-84: *The integrated model framework handles the input datasets required for the PM and GAS modules and develops the LUR models of each air pollutant independently. The LUR models and the corresponding spatial distribution maps within each module can be used to further validate the LUR models and the corresponding spatial distribution maps under another module (Li et al., 2021). For instance, in high-density cities, the spatial distribution of $O_3$ shows a generally opposite spatial variability compared with traffic-related air pollutants because of the $NO_x$ reaction with $O_3$. The established LUR models of the PM and GAS modules were used for the assessment of exposure in epidemiological studies.*

3. **Regarding these comments "The models were specifically developed for "high-density cities". Still, it appears that there is a mismatch between the sparse density of monitoring stations and the actual population density. Could the author provide further clarification on this discrepancy?" and "Overall, it appears that the number of data points presents a significant limitation in this study. Would the author consider adopting a spatio-temporal model to incorporate a larger data set with finer temporal resolution?" The authors responded by saying that the monitoring stations are diverse and representative. Please consider strengthening the discussion on the adequacy of monitoring for this purpose in this city and other high-density cities, and on possible alternative methods that could be used depending on the density of monitors.**

**Response:** It should be noted that the adequacy of monitoring for the purpose of developing a LUR model should not be determined by the number of stations alone. Some more factors should also be considered, such as whether the stations can reflect the characteristics of the air pollutants at their locations and in their surrounding areas, etc. This is the reasons why we highlighted the representation of the stations in our responses before. Furthermore, we have done detailed model evaluations as reported in our manuscript that provided sufficient evidence showing the adequacy of monitoring for our model development. As suggested by the editor, we strengthened the corresponding discussion.

Line 261-263: *High-density cities usually have spare air quality monitoring stations. This discrepancy clearly highlighted the need to develop LUR models for the spatial mapping of air pollution in high-density cities. This work developed an integrated model framework for a high-density city based on the air quality data collected at the sparse monitoring stations.*

*Line 265-268: Similar to other LUR model studies, one limitation of this study is typically the limited number of monitoring stations. It should be noted that the adequacy of monitoring should not be determined by number of stations alone. This study therefore performed detailed evaluations to examine the adequacy.*

*Line 272-276: The detailed evaluation results have proved that our models had promising performance and are capable of reflecting the air quality characteristics of the city. Therefore, our models are considered as sufficient for the scope of this study. Certainty, it is strongly recommended to carry out a further study using different modelling methods (e.g., machine learning) when more data are available with at a finer temporal resolution or collected from a larger number of monitoring stations.*

**4. Line 141: it's still not clear to me what the "pre-defined" direction refers to. Where is it defined?**

**Response:** The "pre-define" is indeed confusing. We revised the related content to make it clearer (Line141-143, Line 147).

*Line 141-143: The method computed the direction of effect for a predictor variable to reflect the effect of the predictor variable on air pollutant concentration. While the effect of a predictor variable can be positive or negative, the direction of effect for each type of predictor variable was first judged based on the currently known relationship between the predictor variable and the corresponding air pollutant. Line 147: ... the direction of effect was consistent with our judgement.*

**5. Line 128: "To include the effect of potential transboundary pollution, the geo-locations of longitude and latitude were also adopted which could reveal a north-south or west-east gradient of air pollutant concentrations (Huang et al., 2017)." I don't think this statement is correct. The longitude and latitude are associated with north-south and east-west variability not captured by the other covariates in the model.**

**Response:** We have revised the mentioned content as suggested by the reviewer (Line 133-135).

*Line 133-135: The geo-locations (longitude and latitude) were also adopted because they can reveal a north-south or west-east variability gradient of air pollutant concentrations which are not captured by the selected predictor variables in the model (Huang et al., 2017).*

**6. Line 277: "However, as revealed by the present study and previous studies (HKEPD, 2022; Zeng et al., 2022), O3 pollution has become an emerging issue, especially in rural areas of Hong Kong, which cannot be accomplished through the control of vehicular emissions." I don't think this is quite correct either. Vehicular emission control could feasibly reduce NOx emissions, which may produce O3 that is transported to rural areas. In addition, NOx contributes to HONO, which can be transported to rural areas and lead to O3 formation: [Song, M., Zhao, X., Liu, P. et al.**

**Atmospheric NOx oxidation as major sources for nitrous acid (HONO). npj Clim Atmos Sci 6, 30 (2023).]**

**Response:** We have revised the mentioned content (Line 289-299).

Line 289-299: *For instance, the Hong Kong government has spent tremendous efforts on the reduction of vehicular emissions over the past two decades, which successfully reduced traffic-related air pollutants like PM2.5 and NO2. However, as revealed by the present study and previous studies (HKEPD, 2022; Zeng et al., 2022), O3 pollution has become an emerging issue, especially in rural areas of Hong Kong. The relationship between the control of vehicular emissions and O3 pollution is complex (Song et al., 2023; Zeng et al., 2022). In Hong Kong, NOx reductions from the control of vehicular emissions may lead to an increase in the levels of oxidants, and then cause a net O3 production. It is suggested that the control of volatile organic compounds should be implemented to better mitigate O3 pollution in HK (Zeng et al., 2022). This highlights the importance to simulate multiple air pollutants together during exposure assessments. In addition, more research should be conducted to understand the complex and varying interaction of emission sources, pollutant sensitivity to precursors, and air quality in a city to formulate more effective and specific air quality management policies.*